# Protocol for Enrichment of Murine Cardiac Junctional Sarcoplasmic Reticulum Vesicles for Mass Spectrometry Analysis

**DOI:** 10.3390/ijms26178602

**Published:** 2025-09-04

**Authors:** Chiara Di Antonio, Chiara Marabelli, Rossana Bongianino, Silvia G. Priori

**Affiliations:** 1Department of Molecular Medicine, University of Pavia, 27100 Pavia, Italy; chiara.diantonio01@universitadipavia.it (C.D.A.);; 2Laboratory of Molecular Cardiology, IRCCS ICS Maugeri, 27100 Pavia, Italy; rossana.bongianino@icsmaugeri.it; 3Centro Nacional de Investigaciones Cardiovasculares Carlos III (CNIC), 28029 Madrid, Spain

**Keywords:** sarcoplasmic reticulum, membrane protein, proteostasis, mass spectrometry, cardiac pathologies

## Abstract

The junctional sarcoplasmic reticulum (jSR) is a critical organelle in cardiomyocytes, regulating calcium homeostasis and Excitation–Contraction Coupling (ECC). A quantitative understanding of its protein composition is essential for investigating cardiac physiology and related pathologies. However, isolating intact jSR vesicles, particularly those enriched in membrane proteins, remains a challenging task. Here, we describe our optimized protocol for reproducible enrichment of jSR vesicles from a single murine heart, without the use of antibodies. The protocol enables the recovery of low-abundance membrane proteins while preserving their native interactions with partners. This strategy facilitates the straightforward identification by Mass Spectrometry of highly relevant yet challenging jSR proteins, including the cardiac Ryanodine Receptor and calsequestrin. Our protocol provides a robust tool for studying the structural and stoichiometric organization of the cardiac jSR components in a widely used animal model.

## 1. Introduction

### 1.1. The Pathophysiological Relevance of the Cardiac jSR Compartment

The terminal portion of the sarcoplasmic reticulum (SR), namely the junctional SR (jSR), is a highly specialized compartment enriched in Ca^2+^, crucial for the function of skeletal and cardiac myocytes [1,2]. Different from the longitudinal SR (lSR), which extends along the entire length of the sarcomeres, this specialized region is only found at the sarcomere Z-lines, in physical proximity (less than 20 nm) to the sarcolemmal invaginations, known as T-tubules [1,3,4], and mitochondria [5,6]. As the action potential is uniformly and quickly propagated throughout the entire volume of the cell by T-tubules, it triggers the coordinated release of Ca^2+^ ions from the jSR into the cytosol, and thus synchronized myofilament contraction [7]. This process, known as Excitation–Contraction Coupling (ECC), is initiated by the voltage-gated calcium channels (L-type Ca^2+^ channels, i.e., Ca_V_1.2) in the T-tubules, which open in response to membrane depolarization [2,7]. In the cardiac setting, the local Ca^2+^ concentration increases in the narrow dyadic space between the T-tubule and the jSR, which in turn opens the nearby RyR2 Ca^2+^ channels (cardiac Ryanodine Receptor) [7,8].

RyR2 is organized in clusters so that Ca^2+^ release events can be amplified and coordinated, spatially and temporally, to generate an efficacious cytosolic Ca^2+^ transient. Overall, the complex composed of a RyR2 cluster and a related modulator protein membrane is named the Ca^2+^ release unit (CRU). Along with those proteins directly modulating RyR2 activity, the proteome machinery of the jSR also supports the high-capacity storage of Ca^2+^ ions [9]. Of the luminal Ca^2+^ binding proteins, while cardiac calsequestrin (CASQ2) is known to be a jSR-specific component, the histidine-rich Ca^2+^-binding protein (HRC) has been found either in proximity to RyR2 or the sarco(endo)plasmic reticulum Ca^2+^-ATPase (SERCA) pumps for Ca^2+^ reuptake, distributed along the lSR membrane [10].

The precise spatial organization of these proteins at the nanometer scale, and more critically of the CRU machineries, allows the highly coordinated and rhythmic contractions of the heart [11]. Within the CRU, the single-pass transmembrane proteins Tradin (TRDN) and Junctin (JNT) [12,13], and the luminal CASQ2, are known to modulate RyR2 opening and refractoriness. Non-sense or missense mutations in any of these CRU components affect ECC. Among the multiple hereditary diseases associated with genetic mutations of the cardiac CRU components, Catecholaminergic Polymorphic Ventricular Tachycardia (CPVT) is one of the most severe. The bidirectional/polymorphic ventricular tachycardia typical of CPVT is triggered by emotional stress or physical exercise in patients with a structurally normal heart [14,15,16,17]. Since the first identification of a CPVT causative mutation in RyR2 in 2001 [18], dozens of mutations in proteins of the CRU have been discovered and associated with the disease. The three most prevalent forms of the pathology are distinguished according to the mutated gene: either RYR2, CASQ2, or TRDN. The dominantly inherited CPVT type 1, related to mutations of the RYR2 gene, accounts for up to 70% of CPVT cases [19]. CPVT type 2 is caused by a genetic defect on CASQ2 [20], and can show either a dominant or recessive inheritance [21]. CPVT type 5 is the rarest form. It is recessively inherited and caused by the loss of function of TRDN [22]. All CPVT forms feature Ca^2+^ overload within the jSR and increased RyR2 permeability to Ca^2+^. The most accredited hypothesis is that Ca^2+^ leakage from a few RyR2 channels triggers spontaneous Ca^2+^ release from the neighboring ones in a positive-loop cascade. This, in turn, leads to non-electrically triggered contraction of the cardiomyocyte, degenerating to ventricular arrhythmia (VT) [17].

### 1.2. A Protocol for the Enrichment of Cardiac jSR Vesicles in Their Native State

Despite the extreme clinical relevance of the jSR compartment, its molecular composition and organization are poorly described. Among the technical difficulties that the study of this specialized domain poses, its isolation and separate characterization remain one of the most limiting challenges. The relatively high proportion of transmembrane proteins with respect to membrane lipids [23], the high sensitivity to subtle ionic variations in the soluble protein machinery [21], and its architectural complexity at the boundary between T-tubules and mitochondria [5,6] complicate the procedures for enrichment of the jSR components and the maintenance of protein complexes in their native form.

In this study, we present an enrichment protocol to obtain jSR vesicles from a single wild-type (WT) murine heart. This method enables the efficient recovery of intact membrane proteins in their native state, hence without the need for detergents (often required to solubilize such proteins, but can disrupt native interactions). The robustness of the enrichment and the preservation of the molecular interactions are proven by the reproducible detection of notorious low-abundance membrane proteins, such as RyR2, TRDN, and JNT, and other proteins physically tethered to the jSR compartment, such as JPH2 and Ca_V_1.2. The qualitative MS technique allowed us to profile SR membrane proteins with high sensitivity, overcoming traditional limitations in detecting these hydrophobic proteins. This study aims to demonstrate the protocol’s suitability for proteomic investigations on the structural organization of the jSR.

### 1.3. Interspecies Differences in the Cardiac jSR Compartment

While the protocol here described for the enrichment of native jSR vesicles from WT murine samples is rapidly transferable to any of the multiple murine models of cardiac pathologies, it is imperative to acknowledge the nuances when analyzing other animal models or extrapolating findings to human settings, particularly concerning membrane lipid composition and the composition and stability of CRU components.

More specifically, it is well-established that murine RyR2 is more prone to spontaneous Ca^2+^ release events than human ones, indicating a less stable closed state or higher intrinsic leakiness. Among the other species, it is rabbit RyR2 that, particularly under diastole-like conditions, best replicates the functional attributes of human RyR2 [24]. Notably, these species differences persist even when isolated RyR2 is studied under identical in vitro conditions, suggesting that they are consequences of intrinsic differences in partnering proteins (such as FKBP12.6) [25] and membrane lipid composition [23,26,27]. Known differences in CASQ2 abundance and its vulnerability to loss during isolation explain the significantly lower levels of CASQ2 recovered in SR vesicles in sheep with respect to samples obtained from murine or rat tissues [28]. Regarding the membrane lipid milieu, the heart of small rodents has strikingly more polyunsaturated fatty acids and polyunsaturated cardiolipin acyl chains than larger animals [26,27]. These differences in the membrane fluidity directly affect the open/closure equilibrium of membrane proteins such as SERCA [29] and, by extension, it is expected to influence RyR2 function and clustering.

Beyond the properties of individual proteins, the architecture and protein composition of the CRU widely differ between mice, pigs, and humans. Compared to pigs and humans, rodents feature a higher density of RyR2 and SERCA2 to support their rapid 600+ bpm heart rates, as well as other accessory jSR proteins like junctophilin-2 (JPH2), TRDN, and JNT [30].

In conclusion, while our protocol provides a valuable foundation for murine jSR proteomics, translating these insights to human cardiac ECC necessitates a rigorous consideration of species-specific molecular landscapes. Future investigations leveraging this protocol should diligently account for these potential disparities in membrane composition, protein stability, and quantificational challenges, employing cross-validation with human cellular models where feasible to bridge the translational gap.

## 2. Results

### 2.1. Protocol for jSR Vesicles Preparation

#### Sample Homogenization and Isolation of the Microsomal Fraction

##### Collection of the Heart Tissue

The first step, critical for the quality of the prepared sample, is the collection of the mouse heart (about 100–120 mg of tissue). Hearts are excised from C57BL/6N mice after euthanasia performed by cervical dislocation. Comparison of the results obtained after direct treatment of freshly isolated tissues, or instead freshly fast-frozen in liquid nitrogen, clearly indicates a major degradation of the proteins of interest for the fresh, unfrozen samples (Appendix A). Thus, it is recommended to use fresh tissues that have been immediately fast-frozen in liquid nitrogen just after isolation, as this minimizes degradation processes that may otherwise hamper the enrichment reproducibility and the physiological integrity of the enriched portion. A scheme of the entire protocol for the preparation of jSR vesicles is shown (Figure 1).

##### Homogenization of the Heart Tissue

In total, 1.5 mL of Homogenization Buffer was added to each flash-frozen mouse heart before homogenization with a bead-beater (see Methods Section 4.2.2) [31]. Unlike our enrichment protocol, the standard homogenization procedure involved grinding frozen tissue using a pestle and mortar in liquid nitrogen vapors, followed by resuspension in a Homogenization Buffer and a sonication cycle for a minimum of 4–5 min. This approach introduces several sources of variability, including heat generation, inconsistent lysis efficiency, and potential cross-contamination when multiple samples are processed in parallel. In our case, we opted for an alternative method using the Minilys^®^ bead-beater instrument, which processes each sample individually in a sealed tube and applies a standardized motion that improves reproducibility and reduces contamination risk.

##### Sedimentation of Cellular Debris

The homogenate is subjected to a centrifugation step at 9000× *g* in order to remove cellular debris and major organelles, such as nuclei and mitochondria. Beads are also collected at the bottom of the tube. The supernatant (S) is separated from the rest of the sample and is then filtered with a 100 µm cell strainer for the removal of the fatty layer, composed of unbroken cells or large membrane debris. The filtered supernatant is subjected to ultracentrifugation at 4 °C for 1 h at 200,000× *g* to allow the recovery of cellular material (microsomal fraction) containing SR vesicles. The ultracentrifugation step was incorporated after evaluating different centrifugation protocols. Although published methods [31] recommend 20,000× *g* for the collection of SR vesicles, in our experience, higher sedimentation velocities yielded a more consistent pellet composition and volume, allowing reproducible loading for WB analysis of samples derived from single murine hearts and resulting in clearer detection of jSR protein bands.

##### Quantification of the Protein Content of the SR-Enriched Sample

In the end, the pellet containing SR vesicles is separated from the supernatant and resuspended in 200 µL of Resuspension Buffer. Resuspension with a thin syringe needle and the high ionic conditions prevent membrane aggregation and help maintain the stability of the overall sample. Protein concentration is determined using the Pierce BCA Protein Assay Kit (Thermo scientific, Waltham, MA, USA). To provide a clear evaluation of the enrichment process, we included additional WB analyses at different stages of the preparation (Figure 2; raw images of WB are presented in the Appendix A).

Here, we show the total heart lysate (T), the membrane pellet obtained after the first centrifugation (P), the supernatant from the ultracentrifugation step (S), and the final vesicle pellet (V, Figure 2). To assess the distribution of different cellular components, we used a representative protein from each compartment: mitochondria (MCU), nucleus (Histone H3), and sarcoplasmic reticulum (RyR2 and CASQ2). For each lane, we loaded 30 µg of total protein to ensure a comparable representation of the different fractions. The Western blot analysis here shows samples derived from two independent preparations, meaning that the samples were obtained from two distinct mouse hearts processed simultaneously yet independently.

We recognize that RyR2 is partially lost during homogenization, which is an inherent limitation of the separation process. This loss is primarily due to the mild homogenization conditions we deliberately employ to preserve protein integrity and minimize overall degradation. These conditions result in a fraction of cells remaining intact, thereby retaining some of the RyR2 within non-disrupted cells. Harsher homogenization protocols, while potentially increasing the overall yield, have been observed to cause extensive protein loss (Appendix A). Our approach prioritizes the enrichment of high-quality, CRU proteins, maintaining their native conformations and interactions, which is critical for downstream structural studies.

Regarding the mitochondrial and nuclear proteins, their presence is reduced but not completely eliminated in our final vesicle fraction. While the enrichment factor relative to these contaminants is moderate, it is important to emphasize that the goal of this protocol is not absolute purification but rather an increased representation of CRU-associated proteins compared to total homogenates.

Unlike previous studies, typically starting from larger amounts of tissue obtained from multiple animals [32,33], our approach is optimized for enrichment from a single murine heart (~100 mg). The main limitation to jSR protein enrichment indeed lies in the fact that the jSR compartments in ventricular cardiomyocytes represent the 0.22% of the cellular volume [34] and thus CRU proteins are inherently low in abundance within the total cardiac proteome. A key strength of our protocol is that it balances enrichment with protein integrity, allowing for the isolation of CRU components in a physiologically relevant state from a single murine heart.

### 2.2. Quality Evaluation of the Protocol

#### 2.2.1. Reproducibility Evaluation of the Protocol

To assess the accuracy and reproducibility of the enrichment protocol, we inspected by WB the proteins enriched across nine independent samples, each derived from a distinct mouse heart.

Samples were treated separately and simultaneously. WB analysis of the key proteins involved in ECC (following the protocol described in the Materials and Methods Section 4.2.11) reveals the variance in the protein levels across biological replicates (Figure 3).

The consistent protein expression patterns across biological replicates demonstrate the robustness of this enrichment method in isolating CRU proteins, yielding reproducible results for the soluble protein Casq2, and the small transmembrane proteins Trdn and Jnt, despite the limited starting material (i.e., a single mouse heart). Expectedly, RyR2 exhibits higher variability across biological replicates, reflecting its intrinsic biochemical properties as an MDa-sized transmembrane protein complex with low stability. Indeed, the challenges associated with isolating and maintaining RyR2 channels in their native state due to their large size and poor stability have already been discussed [31,35]. Despite these inherent challenges, our protocol successfully enriches RyR2 in all tested samples, providing valuable insight into the molecular composition of the jSR while preserving key protein interactions.

#### 2.2.2. Mass Spectrometry Analysis (nLC-HRMS)

Three samples, each derived from a single mouse heart, were analyzed by a relatively simple experimental procedure of Mass Spectrometry to conduct a qualitative protein characterization. The samples were analyzed using a nano-LC system connected to an Orbitrap ExplorisTM 240 Mass Spectrometer equipped with a nano-electrospray ion source. Samples were injected in three technical triplicates (see details in Section 4.2.13).

Despite the typical low abundance of RyR2 in the total cellular proteome, our protocol enabled its identification via MS without the need for detergent-assisted extraction. Appendix A lists the identified proteins from the endoplasmic and sarcoplasmic reticulum with a threshold of at least two peptides per hit. The total of the peptides obtained from all three technical replicates was considered.

### 2.3. Fractionation of the SR Vesicles by Density Gradient

Although identification of key jSR proteins is achievable at this level of enrichment, further purification from lSR components can be performed through sucrose gradient fractionation. This approach enables a more refined separation between longitudinal and junctional SR, and allows additional enrichment of the jSR fraction by reducing the abundance of contaminating proteins, such as those from the cytoplasm or plasma membrane. Since the recovery of the gradient fraction is very low, our protocol using jSR vesicle separation employs *n* = 4 murine hearts for a single jSR enrichment.

Sucrose density gradient centrifugation is a gentle method for separating particles in an aqueous solution. The distinct densities of the membranous and protein particles in aqueous solution determine their distribution among differently concentrated sucrose phases when subjected to high centrifugal forces (Figure 4).

Under the effect of acceleration forces toward the tube bottom, each particle or vesicle moves through the sucrose layers until it reaches a point where its density matches that of the surrounding sucrose. Once this equilibrium is achieved, components of different densities, i.e., with distinct lipid and protein composition, will have sedimented at different positions [36].

Previously published density gradients of muscle-derived samples [35,37] already identified vesicles from the longitudinal SR in the 30% (*w*/*v*) sucrose density fractions, whereas vesicles from the junctional SR have been located at the interface between 40% and 50% (*w*/*v*) sucrose phases. Vesicles from the jSR indeed feature a higher protein content with respect to the wet weight of the organelle, whereas longitudinal SR vesicles are characterized by a higher membrane-to-protein proportion [38].

In our case, the jSR protein components display a density-dependent distribution, accumulating primarily at the 40–50% (*w*/*v*) sucrose interface (Figure 5). The fact that the soluble protein CASQ2 follows the same enrichment trend as the membrane-bound CRU RyR2 and TRDN indicates its tight association with them, and hence that the jSR vesicles are preserved under these experimental conditions. Assessment of additional SR proteins is employed as a further confirmation of the quality of the obtained sample. SERCA (a Ca^2+^ ATPase that pumps Ca^2+^ from the cytosol back into the SR), Ca_V_1.2 (an L-type Ca^2+^ channel), and Junctophilin 2 (JPH2), a membrane-bound protein that anchors the plasma membrane to the SR membrane, are also inspected by WB. SERCA is broadly detected across fractions, as expected for this highly abundant SR protein, consistent with its known role in Ca^2+^ reuptake along the length of the sarcomere. In contrast, the Ca_V_1.2 (a Ca^2+^-channel of the T-tubule, known to lie in close proximity with the jSR Ca^2+^-channel RyR2 [8]) and JPH2, which tether jSR and T-tubule membranes, are identified along with RyR2 within the denser sucrose fractions. This provides an additional confirmation of the retained stability of physiological protein–protein interactions of the dyadic structure.

Non-SR elements, such as the sarcomeric contractile fibrils and mitochondria, are also fractionated by the sucrose density gradient. Mitochondrial contaminants, instead, are much larger and denser than SR vesicles due to their double-membrane structure and the presence of mitochondrial proteins and sediment within the 50% (*w*/*v*) sucrose fraction.

#### 2.3.1. Cryo-EM Analysis

Cryo-electron microscopy (Cryo-EM) was employed to evaluate vesicles’ morphology, detect possible contaminants (such as collagen or actin fibers) (see methodological details in Section 4.2.15). Qualitative Cryo-EM imaging shows features that are consistent with RyR2 channels, with their expected shape and massive dimension (27–28 nm) in multiple vesicles [11] (Figure 6).

#### 2.3.2. Relative Quantification of jSR Vesicles

Fraction 5 of the sucrose density gradient, enriched in jSR components, was subjected to MS analysis as previously described (see Section 2.2.2). To evaluate the degree of enrichment, we compared this fraction to three SR-enriched samples, each derived from an individual mouse heart (not subjected to sucrose gradient fractionation). MS data reveal a general reduction in contaminant proteins, such as cytoplasmic and plasma membrane-associated proteins (Figure 7). These results suggest that the sucrose density gradient can serve to further enrich the sample for the jSR subcellular compartment. 

To provide a more detailed view of the enrichment of SR components, we inspected a subset of selected proteins of interest from the jSR and lSR (Figure 8).

According to the data presented in Figure 8, proteins specifically localized to the jSR are clearly enriched in the corresponding sucrose gradient fraction. In contrast, the relative abundance of other SR proteins, such as HRC and Sarcalumenin, is reduced in this fraction.

## 3. Discussion

We here present a protocol for the enrichment of jSR vesicles from a single murine heart, achieving a purity and reproducibility that allow quantification of low-abundance jSR proteins with Mass Spectrometry. Our approach addresses key limitations of traditional methods, such as variability in homogenization and insufficient resolution of vesicle subtypes, resulting in a robust technique for preparing jSR-enriched fractions suitable for downstream proteomic analyses.

Compared to previous protocols, our method introduces critical improvements that enhance preservation of membrane stability. Indeed, as this enrichment protocol employs a single murine heart, it significantly reduces the amount of biological material usually required [39,40].

These vesicles also preserve the structural integrity of jSR proteins, which is a fundamental step for the study of membrane protein complexes. The observed distribution of key SR proteins, such as RyR2, CASQ2, SERCA, TRDN, and JNT, aligns with their known localization in junctional and longitudinal SR subdomains, validating the efficiency of the protocol in maintaining protein co-assembly and native membrane associations.

We expect this protocol will set the basis for future studies to address the characterization of the SR or jSR proteome across different animal models, either for a comparison of the physiological scenario across different species, or to identify alterations associated with specific genetic or physiological conditions.

Integrating this protocol with more advanced MS and imaging techniques may help uncover novel protein interactions and regulatory pathways within the jSR. Such investigations could ultimately lead to the identification of novel therapeutic targets among these jSR resident proteins.

## 4. Materials and Methods

### 4.1. Materials

Minilys^®^ beads-beater (Bertin technologies, Montigny-le-Bretonneux, France)2.8 mm zirconium oxide beads (Bertin technologies)2 mL reinforced tubes (Bertin technologies)HERMLE Z 216 MK mini centrifuge100 µm cell strainerOPTIMA MAX-XP Beckman Coulter; TLA-120.2 rotorHamilton syringe 500 µLPierce BCA Protein Assay Kit (Thermo scientific)OPTIMA XPN 90; SW 41 Ti rotorUltra-clear centrifuge tubes (14 × 89 mm) (Beckman Coulter, Brea, CA, USA)Polycarbonate centrifuge tubes (11 × 34 mm) (Beckman Coulter)Primo multiwell plate 96 well, flat bottom (Euroclone, Milan, Italy)NanoQuant spectrophotometer (infinite F200 pro, Tecan, Männedorf, Switzerland)Laemmli sample buffer 4× (BIORAD, Hercules, CA, USA)Mini-PROTEAN TGX Stain-Free Gels (BIORAD)Trans-Blot Turbo Transfer Pack (Mini format 0.2 µm PVDF) (BIORAD)Trans-blot turbo (BIORAD)10× TBS (BIORAD)Tween 20 (BIORAD)Precision Plus Protein Standards (Dual Color) (BIORAD)Homogenization Buffer:
0.5 mM EDTA20 mM Na_4_O_7_P_2_20 mM NaH_2_PO_4_1 mM MgCl_2_10% (*w*/*v*) sucroseEDTA-free protease inhibitors (Sigma Aldrich, Saint Louis, MO, USA)
Sucrose-Phase buffers:
0.5 mM EDTA20 mM Na_4_O_7_P_2_20 mM NaH_2_PO_4_1 mM MgCl_2_20–25–30–40–50% (*w*/*v*) sucrose
Resuspension Buffer:
0.5 mM EDTA20 mM Na_4_O_7_P_2_20 mM NaH_2_PO_4_1 mM MgCl_2_10% (*w*/*v*) sucrose400 mM KClEDTA-free protease inhibitors (Sigma Aldrich)
Dilution Buffer:
0.5 mM EDTA20 mM Na_4_O_7_P_2_20 mM NaH_2_PO_4_1 mM MgCl_2_400 mM KClEDTA-free protease inhibitors (Sigma Aldrich)


### 4.2. Methods

#### 4.2.1. Animal Studies

All animal studies were conducted in compliance with the EU Directive 2010/63/EU and according to the Committee for animal well-being of the University of Pavia.

The animal study protocol was approved by the Italian Ministry of Health; protocol code 223/2023-PR; date of approval 17 March 2023.

#### 4.2.2. Tissue Preparation

Flash-freeze the tissue (mouse heart) by placing it in a reinforced tube already containing 6 zirconium oxide beads and then rapidly place it into liquid nitrogen.Extract the tube after at least 5 min and place it on ice.Homogenize for 4 cycles of 30 s (with 30 s of pause in between on ice) at 5000 rpm using a Minilys^®^ beads-beater in 1.5 mL of Homogenization Buffer. Tubes shall be filled to their maximum volume to minimize the formation of air bubbles, which could compromise tissue integrity during homogenization.

#### 4.2.3. Centrifugation

Centrifuge homogenate (including the beads) in a pre-cooled mini-centrifuge at 4 °C at 9000× *g* (HERMLE Z 216 MK mini centrifuge) for 20 min.Pipette the supernatant through a 100 µm cell strainer positioned on top of a 50 mL Falcon, positioned on ice.

#### 4.2.4. Ultracentrifugation

Move the filtered supernatant to a 1.5 mL polycarbonate centrifuge tube and ultracentrifuge it at 4 °C for 1 h at 200,000× *g* (Beckman Coulter OPTIMA MAX-XP, TLA-120.2 rotor).

#### 4.2.5. Pellet Resuspension

Remove about 1 mL of supernatant with a 200 µL micropipette, without touching the pellet. Always keep the sample on ice.Resuspend the remaining pellet carefully in 200 µL of Resuspension Buffer using a 500 µL Hamilton syringe.

#### 4.2.6. Protein Quantification

To determine the protein concentration of each sample using Pierce BCA Protein Assay Kit (Thermo Scientific), mix 5 µL of sample with 5 µL of SDS 10% (*w*/*v*) in a dedicated 0.5 mL tube.Prepare a serial 1:1 dilution of albumin (BSA) standards, starting from a maximum concentration of 2 mg/mL.Prepare BCA working reagents (WR) according to manufacturer instructions: Use the following formula to determine the total volume of WR required for the assay: (# standards + # unknowns) × (# replicates) × (volume of WR per sample) = total volume WR required.Prepare WR by mixing 50 parts of BCA reagent A with 1 part of BCA reagent B (50:1, Reagent A:B).Put 190 µL of WR and 10 µL of BSA standards and sample into a 96 multiwell plate, flat-bottom.Cover the plate and incubate at 37 °C for 20 min.Set the spectrophotometer (NanoQuant) to 595 nm absorbance.Prepare a standard curve by plotting the 595 nm measurement for each BSA standard vs. its concentration in µg/mL. Use the standard curve to determine the protein concentration of each unknown sample.

#### 4.2.7. Sucrose Density Gradient

Starting from the 50% (*w*/*v*) sucrose buffer, layer with a P200 micropipette 1 mL of each Sucrose-Phase Buffer in decreasing sucrose percentage order (i.e., the order is 50%, 40%, 30%, 25%, and 20% (*w*/*v*) sucrose). Each sucrose-containing buffer has to be layered with care, avoiding mixing with other phases, in an ultra-clear centrifuge tube. The interfaces between the phases should be visible under a source of light.The sample (previously resuspended in 10% sucrose buffer, as specified for Resuspension Buffer) is layered on top of the sucrose gradient.Ultracentrifuge at 4 °C for 1 h at 100,000× *g* in a swinging-bucket rotor (OPTIMA XPN 90; SW 41 Ti rotor).

#### 4.2.8. Fractionation

Fractionate the sucrose density gradient by carefully removing successive layers from the top of the tube with a micropipette. The volume of each fraction is 1 mL for a total of N = 6 fractions. Position each fraction within a 1.5 mL polycarbonate centrifuge tube.

#### 4.2.9. Sucrose Removal

Ultracentrifuge all fractions at 4 °C for 1.5 h at 200,000× *g* using an OPTIMA MAX-XP ultracentrifuge (TLA-120.2 rotor) in a polycarbonate centrifuge tube. This step is critical because residual sucrose could interfere with downstream processes or analyses, such as Mass Spectrometry.

#### 4.2.10. jSR Vesicles Resuspension

Remove the supernatant of all fractions with a micropipette, without touching the pellet.Resuspend the resulting pellet in 200 µL of Dilution Buffer with a 500 µL Hamilton syringe on ice. Samples can be flash-frozen in liquid nitrogen and then stored at −80 °C, before further analysis.The total protein content from each fraction is measured using the Pierce BCA Protein Assay Kit (Thermo Scientific). The first and last fractions of the gradient (the lightest contains 10% of sucrose; the heaviest contains 50% of sucrose) should be devoid of proteins, and thus can be discarded after protein quantification.

#### 4.2.11. SDS-PAGE and Western Blot

Take 30 µg of the total protein sample. Mix with Laemmli Sample Buffer with 1 part of buffer and 3 parts of sample, then incubate at 90 °C for 5 min.Run the acrylamide precast gel until the dye front reaches the reference line.Place the PVDF Mini membrane and bottom stack on the cassette base for the transfer.Place gel on top of the membrane.Place the second wetted transfer stack on top of the gel.Close and lock the cassette lid and insert it into the instrument (Trans-blot Turbo, BIORAD, Hercules, CA, USA) and begin transfer with the following program: 25 limit (V); 1.3 const (A); 7 time (min).Carefully transfer the membrane to a suitable container to proceed with the following steps.Leave the membrane for 1 h with 5% milk in Tris-buffered saline with 1% of Tween 20 (TBS-T) (*w*/*v*).Wash the membrane at least 3 times, 5 min each, with a suitable amount of TBS-T buffer.Incubate the membrane with the desired primary antibody. We suggest, especially for the CRU primary antibody, overnight incubation for a good resolution.Wash the membrane with TBS-T for 2 h, changing the buffer at least 3 times.Incubate the membrane with the secondary antibody for 1 h.Wash the membrane with TBS-T for 1 h, changing the buffer at least 3 times.

#### 4.2.12. MS Sample Processing and Digestion

The sample (30 μg of proteins, quantified with Pierce BCA Protein Assay Kit Thermo Scientific), according to the procedure provided in Section 4.2.6, was processed with 45.5 μL of 50 mM ammonium bicarbonate. After confirming that the pH was basic, with pH test strips, digestion was carried out by adding the following:A total of 3 μL of 100 mM dithiothreitol (DTT) (final concentration: 5 mM), incubated at 55 °C for 30 min.A total of 6 μL of 150 mM iodoacetamide (IAA) (final concentration: 5 mM), incubated in the dark for 20 min.
NB. Excessive IAA may affect the subsequent step of trypsin digestion, and hence any variation in the amount of IAA used may keep this effect into account.
A total of 5 μL of 0.2 μg/μL trypsin, incubated at 37 °C overnight.A total of 1 μL of 100% trifluoroacetic acid (TFA).

After digestion, the sample was purified using a ZipTip (5 μg capacity) as follows:A total of 35 μL of the peptide mixture;ZipTip purification according to the manufacturer’s protocol;SpeedVac centrifugation at 30 °C;Elution in 20 μL of 0.1% formic acid.

#### 4.2.13. MS Measurement

The samples have been analyzed at UNITECH OMICs (University of Milano, Milan, Italy) using a Dionex Ultimate 3000 nano-LC system (Sunnyvale, CA, USA) connected to an Orbitrap ExplorisTM 240 Mass Spectrometer (Thermo Scientific, Bremen, Germany) equipped with a nano-electrospray ion source. Peptide mixtures were pre-concentrated onto an Acclaim PepMap 100—0.3 × 5 mm C18 (Thermo Scientific) and separated on an EASY-Spray column ES902, 25 cm × 75 μm ID packed with Thermo Scientific Acclaim PepMap RSLC C18, 3 μm, 100 Å using mobile phase A (0.1% formic acid in water) and mobile phase B (0.1% formic acid in acetonitrile 20/80, *v*/*v*) at a flow rate of 0.300 μL/min. The temperature was set to 35 °C, and samples were injected in three technical triplicates. The injection volume was 5 μL.

One blank was run between each technical replicate to prevent sample carryover. MS spectra were collected over an *m*/*z* range of 375–1500 Da at 120,000 resolution, operating in the data-dependent mode, with a cycle time of 3 sec between master’s scans. HCD was performed with a collision energy set to 35 eV. Polarity: positive.

#### 4.2.14. MS Data Processing and Evaluation

Three technical replicates of the same sample were processed with the software Proteome Discoverer 2.5 with the database Mus musculus (sp_tr_incl_isoforms TaxID = 10090_and_subtaxonomies) (v2024-10-02).

Filters applied to the analysis were as follows:

Dynamic Modifications:Max. Equal Modifications Per Peptide: 3.Max. Dynamic Modifications Per Peptide: 4.Dynamic Modification: Oxidation/+15.995 Da (M).

Static Modifications:Static Modification: Carbamidomethyl/+57.021 Da (C).

Dynamic Modifications (protein terminus):N-Terminal Modification: Acetyl/+42.011 Da (N-Terminus).N-Terminal Modification: Met-loss/−131.040 Da (M).N-Terminal Modification: Met-loss + Acetyl/−89.030 Da (M).

Filters applied to the results are as follows:Protein level: peptide ≥ 2.Peptide level: Xcorr ≥ 2.2; Rank = 1; Confidence = high.PSMs level: Xcorr ≥ 2.2.

#### 4.2.15. Cryo-EM Imaging

A 3 μL drop of concentrated sample (fraction 5 of the sucrose gradient after buffer exchange) was deposited onto a QuantiFoil copper R2/2 grid and plunge-frozen in liquid ethane. The sample was imaged with a FEI, Talos Artica 200 kV equipped with a FEG and a Falcon 3 camera (FEI).

## Figures and Tables

**Figure 1 ijms-26-08602-f001:**
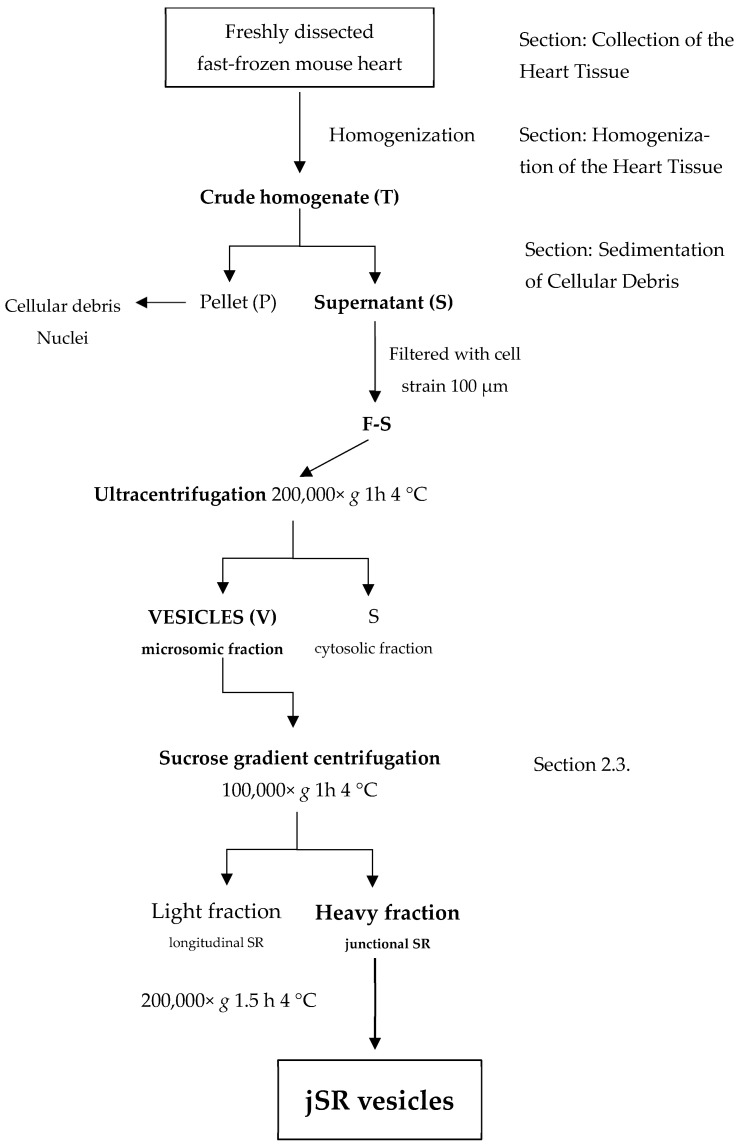
A schematic visualization of the enrichment protocol.

**Figure 2 ijms-26-08602-f002:**
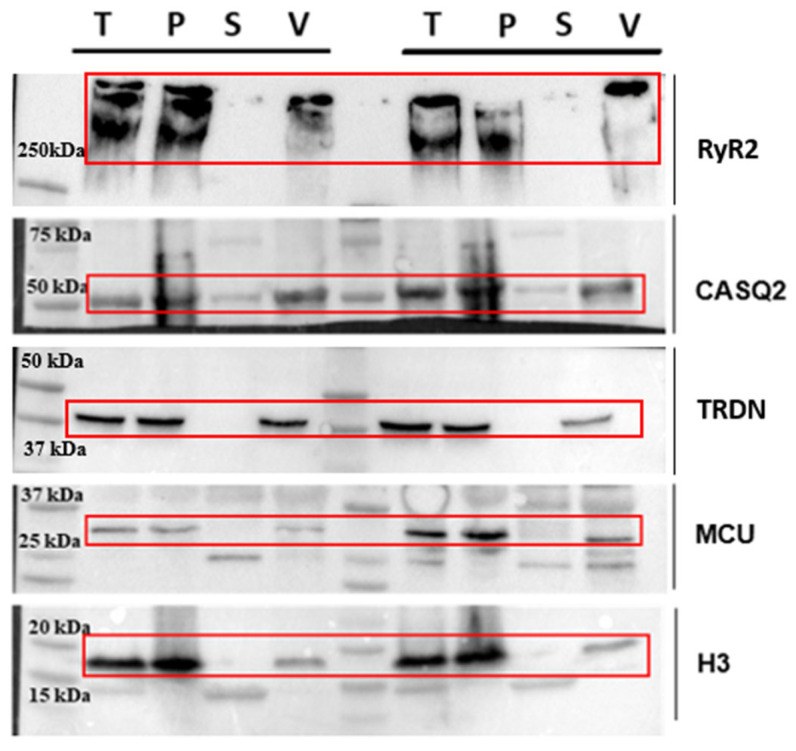
Western blot analysis performed at different stages of preparation. Analyzed steps include total heart lysate (T), membrane pellet (P) obtained after the first centrifugation, supernatant (S) of the ultracentrifugation step, and final vesicle pellet (V) obtained after ultracentrifugation. The antibodies used are sarcoplasmic reticulum proteins RyR2, CASQ2, TRDN; mitochondrial protein MCU; and nuclear protein H3. Red frames indicate the specific band for each protein.

**Figure 3 ijms-26-08602-f003:**
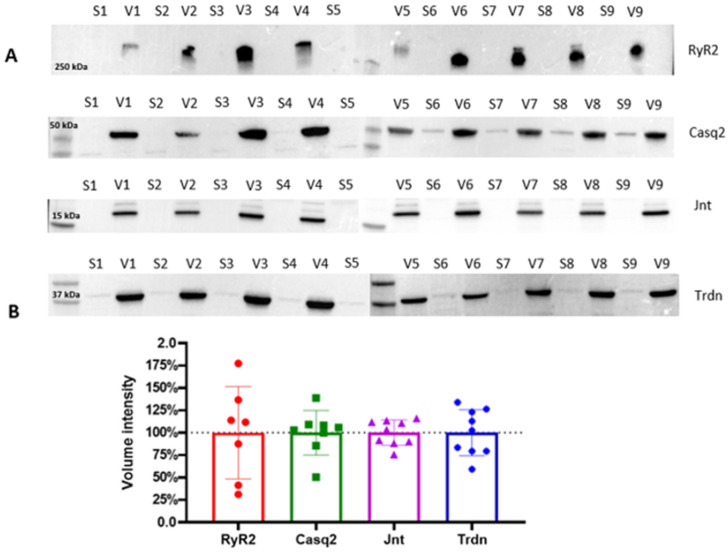
(**A**) Western blot analyses of vesicle fractions (V) obtained from the final pelleting step, compared to the relative discarded supernatant (S). Key jSR and SR proteins, including RyR2, Casq2, Jnt, and Trdn, are analyzed across each preparation. The four horizontal blot panels correspond to different molecular weight protein targets, as indicated by the molecular weight markers on the left (250 kDa, 50 kDa, 35 kDa, and 17 kDa). (**B**) Quantitative analysis of the Western blots is presented in the form of bar plots, summarizing the statistical variance in protein levels across the nine biological replicates. Band intensities were quantified and normalized, and data are represented as mean ± standard deviation. Dotted line represents the mean total volume intensity for each protein.

**Figure 4 ijms-26-08602-f004:**
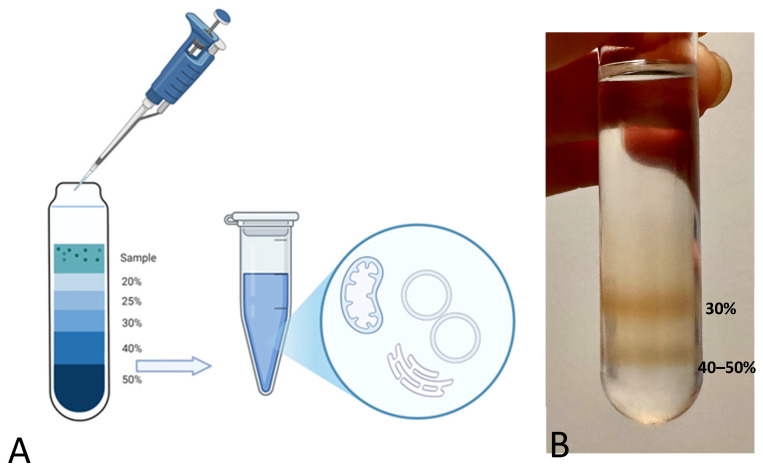
(**A**) Schematic representation of the sucrose density gradient. (**B**) Picture of the sucrose density gradient with two evident bands in the bottom sucrose phases. The uppermost band corresponds to the 30% sucrose phase, while the lower band corresponds to the interface between the 40% and 50% sucrose densities.

**Figure 5 ijms-26-08602-f005:**
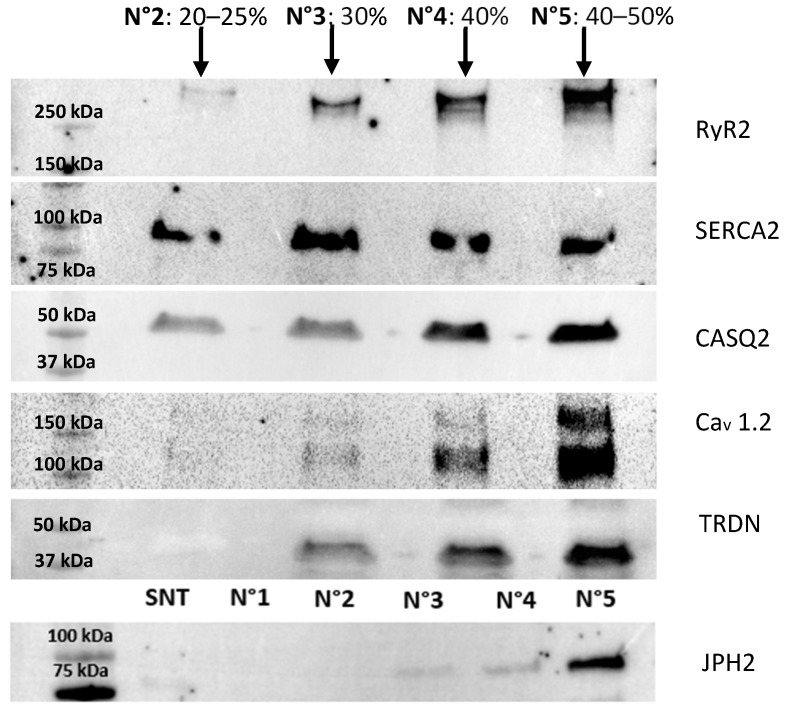
Western blot analysis of fractions 2, 3, 4, and 5 from the sucrose density gradient. The analysis highlights the distribution of proteins across the gradient, with the highest enrichment of the target proteins observed in fraction 5, consistent with the expected localization of SR-derived vesicles containing junctional SR components. These results confirm the efficacy of the isolation protocol in separating SR subdomains and concentrating the proteins of interest in fraction 5. The JPH2 membrane was loaded differently from the others, specifically with the supernatant (S) from the vesicle preparation and fraction N°1, which does not contain any proteins. The total protein content of the fractions is n°2 = 119 µg; n°3 = 182 µg; n°4 = 219 µg; and n°5 = 350 µg.

**Figure 6 ijms-26-08602-f006:**
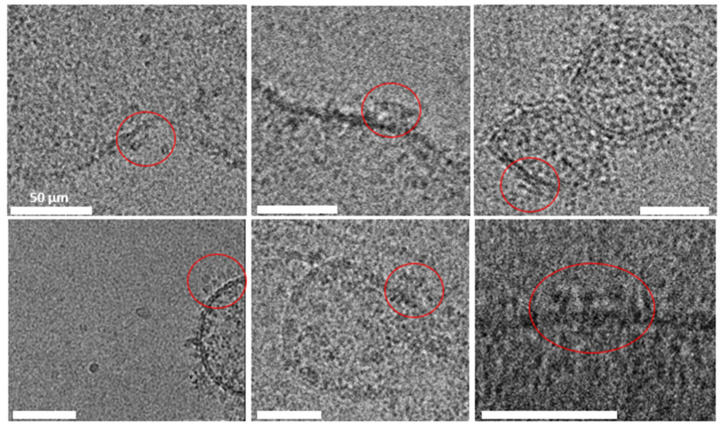
Cryo-electron microscopy images of jSR vesicles. The well-preserved membrane integrity indicates successful isolation and preparation, highlighting the suitability of the protocol for structural studies of jSR-associated proteins. RyR2 channels are circled in red; the scale bar for all pictures is 50 μm.

**Figure 7 ijms-26-08602-f007:**
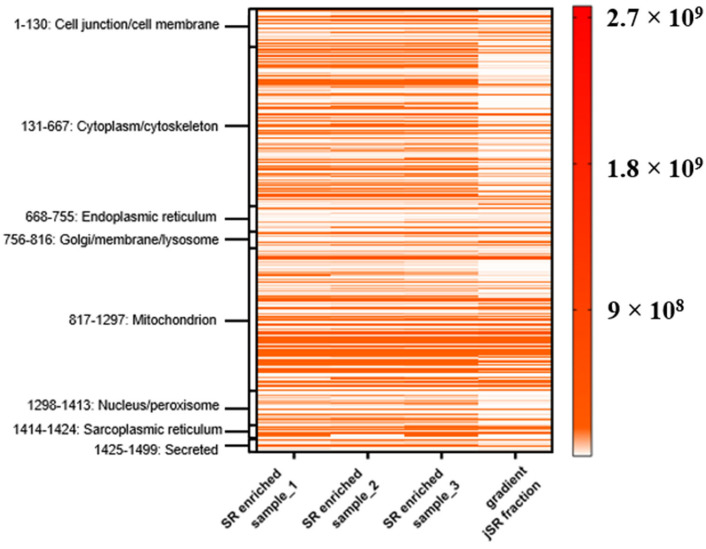
Heat map of protein intensity across three SR-enriched samples (each from *n* = 1 mouse heart) and the sucrose gradient fraction (from *n* = 4 mouse hearts). Proteins were grouped according to their annotated subcellular localization. Each row represents a protein, and the color intensity reflects MS-based quantification (label-free intensities), according to the color legend bar. The fourth column (“gradient”) corresponds to fraction 5 of the sucrose density gradient.

**Figure 8 ijms-26-08602-f008:**
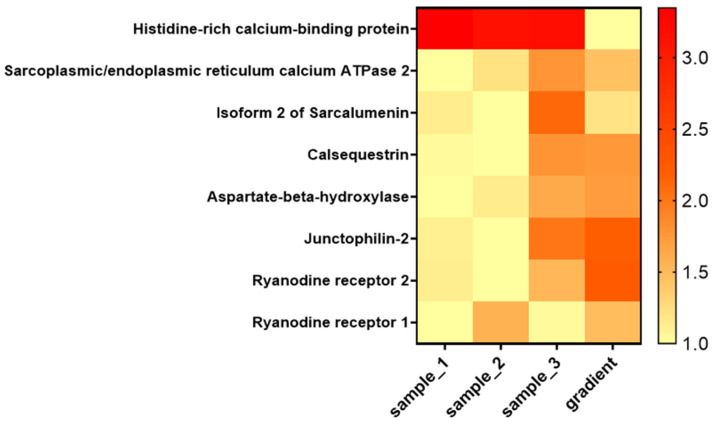
Heat map showing the relative abundance of selected junctional SR proteins across SR-enriched samples and the sucrose gradient fraction. The heat map includes key proteins localized to the jSR and lSR. Values represent intensities from MS analysis normalized over the minimum value.

## Data Availability

Data from this protocol are available upon request to the authors.

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
