# Peer review of "Protocol for Enrichment of Murine Cardiac Junctional Sarcoplasmic Reticulum Vesicles for Mass Spectrometry Analysis"

_ijms, 2025, doi:10.3390/ijms26178602_

Round 1

Reviewer 1 Report

Comments and Suggestions for Authors

Introduction. The text provides an excellent foundation for understanding the role of SR in cardiac physiology. However, it requires additional information on interspecies differences and their impact on disease modeling. Including such data would make the text more relevant to translational research. For instance, studies on "humanized" models, such as mice with human RyR2 or SERCA2a, help minimize interspecies differences.

Results. Discussion:

This section provides a detailed description of the protocol for isolating junctional sarcoplasmic reticulum (jSR) vesicles from mouse hearts but lacks clarifications in some areas.

The protocol uses a sucrose gradient but does not address how differences in jSR membrane lipid composition (e.g., cardiolipin levels) between mice and humans may affect the results.

Additionally, it fails to address how variations in RyR2 stability (RyR2 is more unstable in mice) or CASQ2 buffer capacity (CASQ2 is more susceptible to loss during centrifugation in mice) could impact the transfer of proteomic data to humans.

Check the formulas on line 140 for correctness and the description of the reagents on line 142 and onwards for completeness.

The article needs to be edited to appear more scientific. A detailed description of the protocol and methodological nuances should be included in the supplementary material.

Author Response

We here paste the point-by-point response to the helpful comments of Reviewer 1. Please find the complete responses to both Reviewers in the attached file.

Reviewer 1

Introduction. The text provides an excellent foundation for understanding the role of SR in cardiac physiology. However, it requires additional information on interspecies differences and their impact on disease modeling. Including such data would make the text more relevant to translational research. For instance, studies on "humanized" models, such as mice with human RyR2 or SERCA2a, help minimize interspecies differences.

Results. Discussion:

This section provides a detailed description of the protocol for isolating junctional sarcoplasmic reticulum (jSR) vesicles from mouse hearts but lacks clarifications in some areas.

The protocol uses a sucrose gradient but does not address how differences in jSR membrane lipid composition (e.g., cardiolipin levels) between mice and humans may affect the results.

Additionally, it fails to address how variations in RyR2 stability (RyR2 is more unstable in mice) or CASQ2 buffer capacity (CASQ2 is more susceptible to loss during centrifugation in mice) could impact the transfer of proteomic data to humans.

We sincerely thank the Reviewer for his/her thoughtful suggestion. We agree that addressing interspecies differences is crucial for the translational value of future studies employing our protocol for jSR vesicles enrichment.  This protocol was indeed specifically tailored for murine models, where an abundance of genetically modified lines and disease models are already available in many laboratories investigating excitation–contraction coupling. However, we fully recognize that intrinsic limitations would arise when moving to other animal models, or even human tissues, and caution is needed for extrapolation of results to human cardiac physiology.

Following the reviewer’s suggestion, we now introduced a new subsection in the Introduction (lines 95–126, section 1.3. Interspecies differences in the cardiac jSR compartment), where the following inter-species differences are addressed:

  • The higher intrinsic instability of murine RyR2 compared to human RyR2, and the functional resemblance of rabbit RyR2 to human channels under diastolic conditions.
  • The species-dependent variations in CASQ2 retainment during isolation procedures.
  • Differences in cardiac lipid composition, particularly the higher content of polyunsaturated fatty acids and cardiolipin acyl chains in rodents versus larger mammals, and their impact on SERCA2a and RyR2 function.
  • Broader architectural differences in CRU composition and density between murine, porcine, and human hearts.

We hope that this addition satisfactorily addresses the Reviewer’s insightful comment and strengthens the translational perspective of our work.

Check the formulas on line 140 for correctness and the description of the reagents on line 142 and onwards for completeness.

We thank the Reviewer for his attention to such important details. We now adjusted the typo, now at lines 148-149, as follows :

“20mM Na4P2 O7, 20mM NaH2PO4, 1mM MgCl2 and 10% (w/v) sucrose [20]”.

We also carefully checked other formulas and reagents in the manuscript.

The article needs to be edited to appear more scientific. A detailed description of the protocol and methodological nuances should be included in the supplementary material.

We thank the Reviewer for the suggestion. The “Results” section has been fully revised to remove technical details, and to improve readability. On the other hand the “Methods” section has been adjusted by inserting notes and comments to the rationale of specific precural steps. All amendments are highlighted in yellow.

Reviewer 2 Report

Comments and Suggestions for Authors

The authors report a protocol for enriching jSR vesicles, and suggest that it has advantages over current methods. Apart from using Minilys bead homogenization, the protocol uses a fairly standard centrifugation approach coupled with sucrose sedimentation. Their enriched fraction contains SR proteins, however, even after sucrose fractionation there are still “cytoplasmic and membrane-associated proteins” (L358): what degree of purification was achieved by this protocol and what evidence is there that this an improvement on previous methods?  Further, there is little evidence that these preparations preserve the “structural and functional integrity” (L389) of vesicles or are suitable for “functional analysis” (L384). 

There are a number of other concerns with the manuscript:

  1. L123-126: Are you saying that you have compared direct treatment with fast-frozen tissues? If so, these data should be given in Results. If not, then please add a reference to this comparative study. Fig S1 only shows that the Minilys procedure can cause degradation of RyR2.
  2. L143 onwards: You imply that the standard homogenization procedure is not reproducible and can result in cross-contamination. Is there any evidence for this statement? Also, where is the evidence that Minilys beads “achieve a consistent level of lysis and homogenization”?
  3. L163: Where are the data showing that “our experiments demonstrate … an improved quality in “western blot”.
  4. Figure 2: The raw data for the westerns (blotted for RyR2, MCU, CASQ2 and H3) that are shown in the “original images” file should be included in Supplementary Information, and linked in the text (e.g. as SI figure Sx). Is there a reason for the order of the westerns in Figure 2 (D,C,B,A) : it would make more sense to place them in order molecular weight (D, B, C, A).
  5. L187: “We used markers…”. Actually, there are no independent markers included in the gel. You are relying solely on the specificity of the antibodies to identify the gel positions of each protein. Figure S1 shows the presence of multiple RyR2 immunoreactive bands, which raised the question as to the specificity of the antibody,
  6. L193 and Figure 2: RyR2 is not “partially lost”; it appears to be absent in the V lane for both samples. There is, at best, some light smearing of the stain in V at the same apparent mass as RyR2 in both T and P samples together with some higher molecular weight immunoreactive material present. CASQ2 is present in both V samples.
  7. L205: There is “increased representation of CRU-associated proteins”? Can you estimate the recovery of vesicles? The mitochondrial and nuclear markers are clearly present; what about other proteins? Have you run a Coomassie or silver stain on your gels to estimate total enrichment? Where is the evidence that this method is better than other reported methods?  
  8. Figure 3a: Presumably these are all pellet (P) and supernatant (S) samples. If so, this doesn’t assess the reproducibility or the robustness of the protocol, simply that measured after the initial homogenization and first centrifugation step. Are the preparations used for figure 2 included in figure 3a? If so, which are which?  The running position of the RyR2 samples seems very variable, particularly when compared with the faster running lower molecular weight components.
  9. Figure 3b: I assume that you have scanned the blots using a densitometer, averaged the densities for each protein and then plotted the % difference from mean. This only works if the densitometric response is linear and all samples are within the measurable range. In particular, sample P3 (RyY2) looks overloaded; the response may not be linearly related to that of P1.
  10. L2403 onwards (mass spectrometry). It isn’t clear which fractions of mouse heart were used for mass spectrometry - unfractionated mouse heart, whole heart homogenates, pellet, supernatant, vesicles? Unless these results are from the purified vesicles, then it doesn’t add much to the manuscript.  
  11. Figures 4b and 5: Why are the bands at 30% and 40/50% sucrose coloured red/brown? You are assuming that the 40/50% layer contains the junctional vesicles based on previous reports. However, the markers for vesicles, RyR2 and CASQ2, are also present in the 20%, 30% and 40% sucrose samples. This would imply either that that the 40/50% layer only contains a fraction of the vesicles, or the vesicles are no longer fully intact. The abundance of SERCA2 across all sucrose layers suggests it is dissociated from the vesicles.
  12. L346 and Figure 6 (Cryo-EM analysis): “We confirmed the presence of intact RyR2 channels….”. It would be better to say: Cryo EM shows features that are consistent with RyR2 channels.
  13. L353 onwards, Figures 7 and 8 (mass spectrometrry):I don’t find the heat maps colour scheme (Fig 7) particularly easy to follow. It appears that some SR components are lost in the 40/50% sucrose fraction. There are clearly a number of enriched mitochondrial proteins present in this fraction. How "pure" is this fraction?

Minor points:

  1. The Introduction is rather wordy.
  2. Figure 1 could be moved to the Supplementary Information.
  3. Section 2.1 onwards (Results): the is a lot of methodology in thie section. It should appear with the rest of the methods in the Methods section.
  4. Table 1 could be moved to Supplementary Information.
  5. Supplying the full operating procedure with all laboratory details is helpful, but would be better situated in the Supplementary Information. There is no need to explain basic techniques such as sucrose sedimentation (L295-300 and Figure 4A).
  6. L155, fig 1 and elsewhere: The use of “.” when expressing numbers implies a decimal point in English. Use a “,” to separate large numbers: 20,000 x g instead of 20.000 x g.
  7. L156, Figure 1 and elsewhere: It would be better to use “supernatant” rather than abbreviating it to “SNT”.
  8. L552 (digestion): Normally the excess iodoacetamide is removed before adding trypsin.
  9. L576 onwards: Please add the method that was used to quantify the proteins.
  10. Figure S1: presumably the blot is for RyR2. This should be noted in the legend.

Author Response

We here provide the point-by-point response to the helpful suggestions of Reviewer 2. The complete list of responses to both Reviewers, along with the relative images, is provided in the attached file.

Reviewer 2

The authors report a protocol for enriching jSR vesicles, and suggest that it has advantages over current methods. Apart from using Minilys bead homogenization, the protocol uses a fairly standard centrifugation approach coupled with sucrose sedimentation. Their enriched fraction contains SR proteins, however, even after sucrose fractionation there are still “cytoplasmic and membrane-associated proteins” (L358): what degree of purification was achieved by this protocol and what evidence is there that this an improvement on previous methods?  Further, there is little evidence that these preparations preserve the “structural and functional integrity” (L389) of vesicles or are suitable for “functional analysis” (L384). 

We thank the Reviewer for the thoughtful assessment and constructive suggestions. In response to the concerns about the degree of purification, we added quantitative analyses comparing total homogenate (T) and the final cardiac jSR vesicles (V) obtained from four independent mice. These results and the underlying methodology are detailed in our Response to Comment 7 and in the revised figures/legends, providing an explicit estimate of enrichment and recovery. It is not our aim to demonstrate a higher purity compared to previously described methods. Our primary goal is the enrichment of calcium release unit proteins in their native state and from a single mouse heart, to a sufficient level to be detectable by standard Mass-Spectrometry methods. Because of the inherently different conditions in the amount of starting sample and in the absence of detergents, we cannot directly compare our enrichment data to any other protocol published to our knowledge.

Finally, we acknowledge that we do not show any functional assay in the here presented manuscript. We therefore apologize for our incorrect statement and have accordingly removed the terms referring to “functional integrity” from the main text (see sentences highlighted in yellow at: lines 19, 225, 369, and at lines 400-402, and 416).

There are a number of other concerns with the manuscript:

  1. L123-126: Are you saying that you have compared direct treatment with fast-frozen tissues? If so, these data should be given in Results. If not, then please add a reference to this comparative study. Fig S1 only shows that the Minilys procedure can cause degradation of RyR2.

We thank the Reviewer for pointing out the need for this clarification. To address it, we now added a supplementary figure (Figure S1A), which illustrates the comparison between treatments of fresh samples and fast-frozen tissues, which is referenced to at line 135 in the manuscript. The figure previously included in the Supplementary Information (now Figure S1B) indeed only demonstrated the degradation of RyR2 caused by prolonged homogenization with Minilys.  

  1. L143 onwards: You imply that the standard homogenization procedure is not reproducible and can result in cross-contamination. Is there any evidence for this statement? Also, where is the evidence that Minilys beads “achieve a consistent level of lysis and homogenization”? 

We thank the Reviewer for the opportunity to clarify this point. Conventional homogenization methods, particularly when processing frozen tissue, typically involve manual pre-crushing with mortar and pestle followed by sonication. When multiple frozen samples are processed in parallel using shared tools, there is an inherent risk of cross-contamination. Moreover, probe sonication presents additional challenges when working with very small volumes such as those typically used for individual mouse hearts (≈1.5–2 ml). As the height of he sonicator tip from the bottom of the tube is a key parameter in determining the efficacy of sonication, even very small variations can lead either to excessive heat generation and protein degradation, or to insufficient sonication, and especially to low-reproducibility in the amount of lysed Vs. non-lysed material. Bead-based homogenizers such as the Minilys minimize these issues: each sample is processed individually in a sealed tube, eliminating direct contact between samples and thereby reducing any risk of cross-contamination. All tubes are subjected to the same standardized “precession” motion, which allows a lysis with uniform efficiency among parts of the same sample, and reproducibility among different samples. To address the need for clarification of this choice, we now readapted the text at lines 143-152 as follows:

 “ Unlike our enrichment protocol, standard homogenization typically involves grinding frozen tissue in liquid nitrogen vapours using a mortar and pestle, followed by resuspension in buffer and a sonication cycle of at least 4–5 minutes. This approach introduces several sources of variability, including heat generation, inconsistent lysis efficiency, and potential cross-contamination when multiple samples are processed in parallel. In our case, we opted for an alternative method using the Minilys® bead-beater instrument, which processes each sample individually in a sealed tube and applies a standardized motion that improves reproducibility and reduces contamination risk.”

More specifically for our experience with these samples, we observed that Western blot protein bands across became more consistent after adopting the MiniLys procedure. Also, the average total protein yield approximately doubled compared to our previously used sonication procedure. This improvement likely reflects the fact that samples could be homogenized more efficiently and rapidly, without prolonged exposure to heat that can cause protein degradation. We provide here representative WB of RyR2 and CASQ2 from earlier SR vesicle preparations (corresponding to samples V in the manuscript Figure 1), generated prior to the introduction of the Minilys homogenization step. These examples are shown for transparency only and can be found in the attached file. 

  1. L163: Where are the data showing that “our experiments demonstrate … an improved quality in “western blot”.

We thank the Reviewer for raising this point. Our intention was to highlight that, although some published protocols employ lower centrifugation speeds, in our hands higher sedimentation velocities yielded more reproducible recovery in a consistent pellet volume,  which in turn provided clearer and consistent WB signals with the same amount of loaded proteins. However, we agree that the original phrasing was potentially misleading without supporting quantitative data. We have therefore revised the text to clarify our rationale without implying that direct experimental comparison data are provided.

Revised Manuscript Text (line 161-165):

“The ultracentrifugation step was incorporated after evaluating different centrifugation protocols. Although published methods [31] recommend 20,000 × g for the collection of SR vesicles, in our experience higher sedimentation velocities yielded a more consistent pellet composition and volume, allowing reproducible loading for WB analysis of samples derived from single murine hearts, and resulting in clearer detection of jSR protein bands.”

  1. Figure 2: The raw data for the westerns (blotted for RyR2, MCU, CASQ2 and H3) that are shown in the “original images” file should be included in Supplementary Information, and linked in the text (e.g. as SI figure Sx). Is there a reason for the order of the westerns in Figure 2 (D,C,B,A) : it would make more sense to place them in order molecular weight (D, B, C, A).

 We thank the Reviewer for this helpful suggestion. In the revised version, the original western blot images for Figure 2 are now included in the Supplementary Information. To improve the quality of the figure, we repeated the experiments: we reran the blot for RyR2 to replace the previous one, which had poor quality (see also our response to point 6), and we added a new blot for TRDN. In addition, we have reorganized the order of the panels so that the proteins are now presented according to their molecular weight, which provides a clearer and more conventional presentation.

  1. L187: “We used markers…”. Actually, there are no independent markers included in the gel. You are relying solely on the specificity of the antibodies to identify the gel positions of each protein. Figure S1 shows the presence of multiple RyR2 immunoreactive bands, which raised the question as to the specificity of the antibody,

We thank the reviewer for pointing out this ambiguity. In the original text we used the term “markers” imprecisely. We did not mean independent in-gel position markers, instead we intended to indicate proteins that are well-established residents of specific subcellular compartments. We have revised the manuscript accordingly at lines 186-187: “..we used a representative protein from each compartment.. “

Protein identities in our blots were assigned based on the antibody specificity (antibodies that we and others have extensively used in the laboratory) and verified migration patterns consistent with the expected apparent molecular weight under our electrophoretic conditions. To substantiate this point, we here added for the Reviewer western blots from independent murine samples probed with the same antibodies, showing bands at the same apparent molecular weights. We apologize for the confusing wording in the previous version.

Regarding Figure S1 and the anti-RyR2 signal: multiple immunoreactive bands are frequently observed for this high–molecular-weight channel due to partial proteolysis. Throughout the study we interpreted as RyR2 only the predominant band migrating at the expected high molecular weight, and we did not use additional bands for analysis. This “ interpretation” roots in the decennal experience of our laboratory, where samples with high abundance in RyR2 protein show a consistent, strong signal above te 250 kDa reference (see figure here attached).

  1. L193 and Figure 2: RyR2 is not “partially lost”; it appears to be absent in the V lane for both samples. There is, at best, some light smearing of the stain in V at the same apparent mass as RyR2 in both T and P samples together with some higher molecular weight immunoreactive material present. CASQ2 is present in both V samples.

We thank the Reviewer for his/her precise observation. To improve the quality of our data, we repeated the WB analysis for RyR2, and the updated results are now presented in the revised Figure 2 (also reported here in the attached file ). We apologize for the limitations in the previous version. We take here the occasion to note that the very large molecular weight of RyR2 often hampers efficient entry into gel wells, which can compromise the sharpness and clarity of the signal compared to lower–molecular-weight proteins.

 To strengthen the dataset, we also included an additional SR-resident protein, TRDN, in the revised figure. In line with the Reviewer’s suggestion, we have reorganized the order of proteins in Figure 2 according to their molecular weight, which provides a clearer and more conventional presentation. Finally, as requested, the raw images corresponding to the western blots in Figure 2 are now included in the Supplementary Information. 

  1. L205: There is “increased representation of CRU-associated proteins”? Can you estimate the recovery of vesicles? The mitochondrial and nuclear markers are clearly present; what about other proteins? Have you run a Coomassie or silver stain on your gels to estimate total enrichment? Where is the evidence that this method is better than other reported methods?  

We appreciate the reviewer’s point on the importance of further analyzing the protein enrichment. Following the suggestion, we  show in the attached file a stain-free gel acquisition of an SDS-PAGE of four different mice, comparing total homogenate (T) with the final jSR vesicle preparations (V). For each sample, 30 µg of protein, quantified via BCA assay, was loaded.

 To obtain a more precise estimation of vesicle recovery, we additionally performed WB using five antibodies: three representative of the SR and two representative of the sarcomere and nuclei. Band intensity was quantified by densitometry, allowing us to infer the recovery percentage. Importantly, all four independent preparations displayed comparable values. please find the relative figures in the attached file.

Specifically, for the two SR-resident proteins JPH2 and JNT, a clear enrichment is observed in the vesicle fraction. CASQ2 displayed a recovery of approximately 75% relative to total homogenate. In contrast, the signal for the sarcomeric contaminant α-actinin was completely lost during enrichment, and the nuclear marker H3 was reduced to less than half compared to total homogenate. Taken together, these results demonstrate a consistent and reproducible enrichment of jSR-associated proteins, with a concomitant reduction of non-SR contaminants.

  1. Figure 3a: Presumably these are all pellet (P) and supernatant (S) samples. If so, this doesn’t assess the reproducibility or the robustness of the protocol, simply that measured after the initial homogenization and first centrifugation step. Are the preparations used for figure 2 included in figure 3a? If so, which are which?  The running position of the RyR2 samples seems very variable, particularly when compared with the faster running lower molecular weight components.

We thank the reviewer for bringing this issue to our attention. The samples shown in Figure 3a are not the same as those presented in Figure 2. We acknowledge that there was an inconsistency in the labeling of the samples: in Figure 3a, the samples initially indicated as “P” were in fact the vesicle fractions obtained from the final pelleting step, and not the pellets indicated with the letter “P” in Figure 2. To remove this ambiguity, we have now relabeled the vesicle fractions as “V” in Figure 3a.

Following the reviewer’s note, we also updated the schematic in Figure 1 to clearly define the lettering used throughout the manuscript (P, S, V, T, jSR vesicles)  and revised the text accordingly to ensure consistent terminology. These corrections should now clarify the nature of the samples and prevent any misunderstanding.

Figure 3. A) Western blot analyses of fact the vesicle fractions (V) obtained from the final pelleting step, compared with the relative discarded supernatant (S). Key jSR and SR proteins, including  RyR2, Casq2, Jnt, Trdn, are analyzed across each preparation. The four horizontal blot panels correspond to different molecular weight protein targets, as indicated by the molecular weight markers on the left (250 kDa, 50 kDa, 35 kDa, and 17 kDa). B) Quantitative analysis of the Western blots is presented in the form of bar plots, summarizing the statistical variance in protein levels across the nine biological replicates. Band intensities were quantified and normalized, and data are represented as mean ± standard deviation.

  1. Figure 3b: I assume that you have scanned the blots using a densitometer, averaged the densities for each protein and then plotted the % difference from mean. This only works if the densitometric response is linear and all samples are within the measurable range. In particular, sample P3 (RyY2) looks overloaded; the response may not be linearly related to that of P1.

We appreciate the reviewer’s careful assessment and attention to detail. We have indeed acquired all the images with ChemiDoc MP imaging system (BIORAD) through the software Image Lab 5.1. We have used the automatic settings for the time of exposure and the acquisition is stopped immediately 1/10th of second before any pixel in the image becomes saturated. For this reason, being all the samples in the same gel and exposed at the same time, they are all within the same measurable range and hence can be compared.

  1. L2403 onwards (mass spectrometry). It isn’t clear which fractions of mouse heart were used for mass spectrometry - unfractionated mouse heart, whole heart homogenates, pellet, supernatant, vesicles? Unless these results are from the purified vesicles, then it doesn’t add much to the manuscript.  

We understand the need to better clarify the identity of the fractions used and the meaning of such Mass Spectrometric analysis at this point of the protocol. The samples used for the initial MS analysis are indeed not derived from the final purified vesicles obtained after sucrose gradient ultracentrifugation: they are SR-enriched vesicle preparations obtained prior to this final purification step. The subsequent mass spectrometry analysis presented in the heat maps compares these initial SR-enriched vesicles with the fully purified jSR vesicles obtained after the sucrose gradient, highlighting the differences between intermediate and final fractions.

We foresee that including both datasets will be valuable for other researchers, as it demonstrates that even SR-enriched vesicles collected at earlier stages of the preparation can already provide informative MS readouts, particularly for detecting low-abundance proteins. However, following the Reviewer’s note and to reduce the emphasis on this dataset, we have now moved Table 1 to the Supplementary Information (Table S1).

  1. Figures 4b and 5: Why are the bands at 30% and 40/50% sucrose coloured red/brown? You are assuming that the 40/50% layer contains the junctional vesicles based on previous reports. However, the markers for vesicles, RyR2 and CASQ2, are also present in the 20%, 30% and 40% sucrose samples. This would imply either that that the 40/50% layer only contains a fraction of the vesicles, or the vesicles are no longer fully intact. The abundance of SERCA2 across all sucrose layers suggests it is dissociated from the vesicles. 

We thank the Reviewer for raising these important points. The reddish/brown appearance of the visible bands at the 30% and 40/50% sucrose interfaces reflects concentrated membrane/protein material from cardiac homogenates accumulating at these density boundaries.

Our results do not rely solely on previous reports but directly demonstrate vesicle distribution in Figure 5. Consistent with the literature, the 40/50% interface is enriched in junctional SR vesicles, while lighter fractions (20–40%) contain a higher proportion of non-junctional/longitudinal SR membranes. The detection of RyR2 and CASQ2 in lighter fractions reflects the inherent heterogeneity of SR-derived vesicles, which partition across multiple sucrose layers depending on vesicle size, lipid composition, and protein content. Junctional SR membranes exhibit a relatively uniform proteomic profile centered around the calcium release complex, whereas longitudinal SR membranes are enriched in SERCA2a and phospholamban, and are more variable in composition. This explains why SERCA2a-positive vesicles distribute more broadly across fractions (Cala SE, et al.,  Mol Cell Biochem. 2024. PMID: 36798315).

Importantly, the presence of RyR2 and CASQ2 in lighter fractions should be interpreted as evidence of a continuum of vesicle densities, reflecting mixed-composition patches of SR membranes. Indeed, previous studies have shown that Ca²⁺ loading and RyR2 activity/inhibition can shift vesicles between “light” and “heavy” fractions, underlining the dynamic nature of these preparations (Cala SE, et al.,  Mol Cell Biochem. 2024. PMID: 36798315). The effective enrichment of jSR components in a defined fraction  is also confirmed by Mass-Spectrometry analysis.

We also note that loss of CASQ2 across sucrose gradients is a known issue. For instance, Murphy et al. (2011) reported that crude SR vesicle preparations from sheep cardiac muscle typically recover only ~5% of total CASQ2, and this limitation that also affects murine tissue. Thus, the authors even propose to avoid the sucrose gradient purification to those researchers interested in analysis of CASQ2.

Taken together, these data support that the 40/50% fraction remains the most enriched in junctional SR vesicles, while the presence of proteins across additional layers reflects both the biochemical heterogeneity of SR membranes and the dynamic partitioning behavior of these complexes.

  1. L346 and Figure 6 (Cryo-EM analysis): “We confirmed the presence of intact RyR2 channels….”. It would be better to say: Cryo EM shows features that are consistent with RyR2 channels.

We deeply appreciate the reviewer’s careful assessment. Accordingly, we have modified the text in 301-302 with “Qualitative Cryo-EM imaging shows features that are consistent with RyR2 channels”.

  1. L353 onwards, Figures 7 and 8 (mass spectrometrry):I don’t find the heat maps colour scheme (Fig 7) particularly easy to follow. It appears that some SR components are lost in the 40/50% sucrose fraction. There are clearly a number of enriched mitochondrial proteins present in this fraction. How "pure" is this fraction?

We thank the reviewer for this observation and apologize if the colour scheme of Figure 7 is not immediately intuitive. While we sincerely regret any difficulty it may cause in interpretation, when we carefully tested alternative palettes, including the classic red–blue scale, we actually find that alternatives even reduced readability. We therefore retained the current scheme, as it provides the clearest contrast for our dataset.

Regarding the presence of mitochondrial proteins in the 40/50% sucrose fraction, we acknowledge that some SR components appear less represented in this fraction, while mitochondrial proteins are relatively enriched. However, as this contamination does not reach a sufficient level to affect MS analysis of jSR components, we propose that it may even reflect the retainment of native interactions within our samples, similarly to what occurrs for the co-migration of CaV1.2, a channel from the T-tubule, already found in association with jSR vesicles (Chen et al.,  EMBO Rep. 2020 PMID: 32147968). It is well established that mitochondria are physically and functionally associated with junctional SR membranes through specific tethering complexes, and their co-enrichment in this fraction likely reflects a physiological feature of the preparation rather than an artefact (Boncompagni S, et al., Mol Biol Cell. 2009. PMID: 19037102)

Finally, we note that some SR proteins not enriched in this fraction (e.g., HRC) are not strictly localized to the junctional SR, which explains their broader distribution across sucrose layers. To make this point clearer, we now explicitly mention this property of HRC in the Introduction (lines 46–48) with appropriate referencing.

Minor points:

  1. The Introduction is rather wordy.

We thank the Reviewer for this observation. We have substantially shortened and streamlined the Introduction, focusing only on the most relevant aspects to improve clarity and readability. All points were the text has been rephrased have been highlighted in yellow in the revised manuscript.

  1. Figure 1 could be moved to the Supplementary Information.

We thank the Reviewer for this suggestion. However, as part of the revision we modified Figure 1 to also include the naming of specific samples and to define the lettering used throughout the manuscript (P, S, V, T, jSR vesicles). We believe that Figure 1 provides an essential overview of the entire experimental procedure and greatly facilitates the reader’s understanding of the workflow. In addition, overview figures of this type are commonly retained in the main text of methodological articles, as they guide the reader through each procedural step and serve as an immediate reference while following the protocol. For these reasons, we would strongly prefer to keep Figure 1 in the main text, where it best supports clarity and accessibility of the manuscript.

  1. Section 2.1 onwards (Results): the is a lot of methodology in thie section. It should appear with the rest of the methods in the Methods section.

We thank the reviewer for this valuable suggestion. We have carefully revised the paragraph by reducing the amount of technical detail, while preserving the scientific clarity of the section. Parts that have remained after rephrasing are now highlighted in yellow.

  1. Table 1 could be moved to Supplementary Information.

Following the reviewer’s suggestion number 10, now Table 1 is now in the Supplementary Information and referenced to as Table S1.

  1. Supplying the full operating procedure with all laboratory details is helpful, but would be better situated in the Supplementary Information. There is no need to explain basic techniques such as sucrose sedimentation (L295-300 and Figure 4A).

We appreciate the reviewer’s note on the fact that sucrose sedimentation is a widespread method, already employed for multiple purposes. Following the reviewer’s suggestion number 16, we now reduced the amount of techincal details reported in the main text and left only specific procedure details in the Methods section. Those sentences expliciting the rationale for a given step have been moved to the methods section (see lines 466-467, and 472-477).

  1. L155, fig 1 and elsewhere: The use of “.” when expressing numbers implies a decimal point in English. Use a “,” to separate large numbers: 20,000 x g instead of 20.000 x g.

We appreciate the reviewer’s careful attention to detail. Now we have changed “.” with “,” in all the manuscript.

  1. L156, Figure 1 and elsewhere: It would be better to use “supernatant” rather than abbreviating it to “SNT”.

We thank the reviewer for this suggestion. We specified the abbreviation of supernatant in Figure 1 and we decided to use “S” instead of “SNT” in the figures; Following the reviewer’s suggestion, we now use the term “supernatant“ in the main text to improve readability.

  1. L552 (digestion): Normally the excess iodoacetamide is removed before adding trypsin.

We thank the reviewer for his/her attention to detail. In our protocol, we do not remove iodoacetamide prior to trypsin digestion because it is not added in excess. The amount used is optimized for our sample concentration and corresponds to the stoichiometric requirement for complete alkylation. After incubation for 20 minutes in the dark, the residual iodoacetamide is negligible and does not interfere with the subsequent enzymatic digestion. We now explicit this in the methods section at lines 513-514: 

“Excessive IAA may affect the subsequent step of trypsin digestion, and hence any variation in the amount of IAA used may take this effect into account. “.

  1. L576 onwards: Please add the method that was used to quantify the proteins.

We thank the reviewer for the suggestion. Accordingly, we have specified the method, now at lines 504-505:

“The sample (30 μg of proteins, quantified with Pierce BCA Protein Assay Kit Thermo Scientific) according to the procedure provided at paragraph 4.2.6)”

  1. Figure S1: presumably the blot is for RyR2. This should be noted in the legend.

We thank the reviewer for his/her attention to detail. Now we have corrected the legend of Figure S1B. The new figure caption now reads:

“B) WB analysis of RyR2 of 3 mice hearts differently homogenized with Minilys beads beater. Our results indicate that prolonged bead beating leads to increased fragmentation and loss of RyR2, rather than improved retention in the final vesicle preparation.”

Round 2

Reviewer 1 Report

Comments and Suggestions for Authors

I am satisfied with the work carried out by the authors and have received answers to all my questions. Now I believe the manuscript is ready for publication.

Reviewer 2 Report

Comments and Suggestions for Authors

None